# Dynamic Perviousness Has Predictive Value for Clot Fibrin Content in Acute Ischemic Stroke

**DOI:** 10.3390/diagnostics14131387

**Published:** 2024-06-29

**Authors:** Vania Anagnostakou, Daniel Toth, Gergely Bertalan, Susanne Müller, Regina R. Reimann, Mark Epshtein, Jawid Madjidyar, Patrick Thurner, Tilman Schubert, Susanne Wegener, Zsolt Kulcsar

**Affiliations:** 1Department of Neuroradiology, Clinical Neuroscience Center, University Hospital Zürich, Frauenklinikstrasse 10, 8091 Zürich, Switzerland; daniel.toth@meduniwien.ac.at (D.T.); gergely.bertalan@usz.ch (G.B.); susanne.muller@usz.ch (S.M.); jawid.madjidyar@usz.ch (J.M.); patrick.thurner@usz.ch (P.T.); tilman.schubert@usz.ch (T.S.); zsolt.kulcsar@usz.ch (Z.K.); 2New England Center for Stroke Research, Department of Radiology, University of Massachusetts Chan Medical School, 55 N Lake Ave, Worcester, MA 01655, USA; mark.epshtein@umassmed.edu; 3Institute of Neuropathology, University Hospital Zürich, Schmelzbergstrasse 12, 8091 Zürich, Switzerland; regina.reimann@usz.ch; 4Department of Neurology, Clinical Neuroscience Center, University Hospital Zürich, Frauenklinikstrasse 10, 8091 Zürich, Switzerland; susanne.wegener@usz.ch

**Keywords:** blood clot, permeability, stroke, histology, thrombectomy

## Abstract

Dynamic perviousness is a novel imaging biomarker, with clot density measurements at multiple timepoints to allow longer contrast to thrombus interaction. We investigated the correlations between dynamic perviousness and clot composition in the setting of acute ischemic stroke. Thirty-nine patients with large vessel occlusion (LVO) undergoing mechanical thrombectomy (MT) were analyzed. Patients received a three-phase CT imaging pre-thrombectomy and histopathological analysis of retrieved clots. Clot densities for every phase and change in densities between phases were calculated, leading to four patterns of dynamic perviousness: no contrast uptake, early contrast uptake with and without washout and late uptake. Clots were categorized into three groups based on dominant histologic composition: red blood cell (RBC)-rich, fibrin/platelet-rich and mixed. Clot composition was correlated with dynamic perviousness using the Kruskal–Wallis test and Pearson’s correlation analysis. The dynamic perviousness categories showed a significant difference between fibrin-rich clots when compared to RBC-rich plus mixed groups. The uptake without washout category had significantly fewer fibrin clots compared to the uptake with washout (*p* = 0.036), and nearly significantly fewer fibrin clots when compared to the no uptake category (*p* = 0.057). Contrast uptake with different patterns of contrast washout showed significant differences of the likelihood for fibrin-rich clots.

## 1. Introduction

Since mechanical thrombectomy (MT) became the standard of care in acute ischemic stroke (AIS), increased research interest has been addressed to study clot characteristics with advanced neuroimaging methods. It has been shown that the histological clot structure may affect revascularization results [1,2]. As such, the predictive value of imaging-based characterization of the thrombus has great potential, because it may influence revascularization techniques and approaches [3]. 

The composition of the thrombus largely influences its mechanical properties [4], which in turn, are important factors in determining MT success and good clinical outcome. For example, aspiration is more successful with soft clots, whereas extraction of harder clots should be attempted with stent retrievers [5,6]. The ratio between red blood cells (RBCs) and fibrin has been found as one of the most important determinants of the clot’s mechanical properties [4]. Mechanically retrieved clots show significant diversity, as well as different responses to thrombolysis and thrombectomy procedures: RBC-rich clots are more responsive to revascularization attempts compared to their fibrin-rich counterparts [7]. If the histological and physical clot structure could be assessed prior to intervention, the most appropriate technique for clot removal could be selected, improving the success rate of MT. Therefore, increased research interest has been addressed to predict clot characteristics before intervention. 

Computed tomography (CT) is the most widely used imaging modality to make time-critical decisions, since it is present in most emergency departments. There is evidence, that thrombus properties correlate with neuroimaging markers on CT. For example, the presence of the hyperdense artery sign on non-contrast CT (NCCT) is associated with a higher content of RBCs and it is often absent in fibrin/platelet-rich thrombi [8,9,10]. Shin et al. [1] observed that RBC-rich clots were associated with the presence of a hyperdense artery sign and with successful recanalization. However, several studies have reported contradictory findings [11,12]. According to their observations, higher Hounsfield unit (HU) values were associated with higher fibrin/platelet content (not RBCs) and longer intervention times [12], or showed no correlation with recanalization success or RBC content [10,11]. 

Thrombus perviousness is a relatively new imaging biomarker [13,14]. It is related to the permeability of the blood clot, which determines the amount of partial blood flow through the occluded artery [5,6,7]. It is thought that thrombus with relatively high permeability can slow down tissue damage and increase the time window for MT as oxygenated blood can reach, at least partly, the ischemic brain tissue in the impacted area. In addition, clot permeability may increase the effect of a concomitant tissue-type plasminogen activator and the efficacy of MT [15].

Perviousness aims to describe the interaction of contrast material with the thrombus, as measured on CT-scan. Standard perviousness is described by thrombus attenuation increase (TAI) measured as the mean clot density difference between NCCT and CTA [10,16,17]. In general, perviousness is measured by manually placing spherical regions of interest (ROIs) with a diameter of 1–2 mm on the clot, both on NCCT and CTA. The average HU of every ROI is calculated and used to compute the mean HU on NCCT (ρNCCT) and CTA (ρCTA). Perviousness is then computed as TAI = ρCTA − ρNCCT. While this method only estimates the real permeability of the clot, it is widely accepted for determining clot perviousness in the clinical setting because it is relatively fast and uses only standard CT images. 

Several studies showed that increased thrombus perviousness was associated with better recanalization and/or a better outcome [13,16,18,19,20]. However conflicting results have been also reported, in which no correlation between clot perviousness and clinical parameters were found [21,22]. Dutra et al. [18] observed an association between perviousness and outcome but none with recanalization. Kappelhof et al. [22] observed a better response to thrombolytic treatment in patients with more pervious clots, but permeability had no effect on reperfusion success. Some studies even reported no correlation between perviousness and patient outcome [10]. Furthermore, perviousness seems to correlate with the histological composition of the clot, but the published results are also contradictory. Shin et al. and Hund et al. [1,23] showed that pervious clots had a higher percentage of red blood cells, while impervious clots had a higher fibrin/platelet content. In contrast, several studies reported that permeable thrombi were associated with higher fibrin/platelet content and a lower percentage of RBCs [18,19], and no correlation between perviousness and histological composition [24] or patient outcome was found [13,14,24,25]. 

The conflicting results may be partly related to the use of single-phase CTA and only two imaging time points for standard thrombus perviousness calculation. Thrombus perviousness is related to its porosity and shape as well as to the connectivity of the pores created between fibrin filaments, trapped platelets and red blood cells [26,27,28]. Standard perviousness provides comparison of clot density between NCCT and CTA, giving the difference between only two imaging time points, thus not considering the contrast agent uptake dynamic in the clot (e.g., late contrast agent uptake or early washout), which may be influenced by the clot structure and other flow related characteristics such as clot and artery shape [29]. To avoid the limitation of quantifying thrombus perviousness using only two imaging time points, as it is standard in the literature, we have introduced the concept of dynamic perviousness, considering longer contrast to clot interaction times using and additional late venous time point (CTV) after the CTA acquisition [30]. Our results using a different patient group to this study and three imaging time points highlighted the importance of the late uptake component of contrast agent, which was correlated with clinical parameters. The purpose of this study was to investigate the relationship between clot histologic composition and perviousness. To overcome the pitfalls of using only two imaging time points for perviousness characterization, we used an additional CTV time point after CTA and computed TAI between CTA and NCCT as well as between CTV and CTA. Using three imaging time points, we characterized the late uptake and early washout of contrast agent uptake, which we will call dynamic perviousness in this study. Dynamic perviousness was correlated with RBC and fibrin/platelet content of thrombi determined by histological examination. 

## 2. Materials and Methods

### 2.1. Patient Selection

The study was approved by the regional ethics committee. A retrospective analysis of 475 consecutive patients referred for MT due to LVO in our hospital between 2019 and 2021 was performed. Patients with occlusions of the intracranial internal carotid artery (ICA), proximal middle cerebral artery (MCA) up to the proximal M2 segment and basilar artery were included. Further inclusion criteria were as follows: (1) the availability of pre-intervention CT imaging (NCCT, CTA) with an in-plane resolution below 0.8 mm, (2) the availability of CTV after contrast agent administration, (3) relative low motion artifacts on CT images, and (4) no previous contrast agent administration for another imaging procedure. This resulted in 137 patients.

Out of the 137 patients, histopathological clot analysis was available for 39 patients, who were selected for further analysis. Each of these patients received MT performed with the use of a stent retriever, direct aspiration, or a combination of both. In addition, the size and quality of the removed clot were sufficient for histopathological analysis. 

Revascularization results were assessed with the modified Thrombolysis In Cerebral Infarction (mTICI) scale (ranging from 0 for no revascularization to 2c for near complete and 3 for full revascularization) [31]. Successful recanalization was defined as an mTICI score of IIc or III, implying either near total or total reperfusion, respectively, of the occluded vessel territory.

### 2.2. CT Imaging Protocol and Dynamic Perviousness Calculation 

Figure 1 shows representative CT images of our clinical AIS protocol. CT imaging was performed on a variety of scanners depending on the referring clinic including Siemens Somatom X.cite (*n* = 23), Somatom Definition Flash (*n* = 8), Somatom Definition AS+ (*n* = 4), Somatom Definition Edge Plus (*n* = 2) (Siemens, Erlangen, Germany), GE Revolution (*n* = 1) (General Electric, Boston, MA, USA) and Philips Brilliance iCT 256 (*n* = 1) (Philips, Amsterdam, Netherlands). After the unenhanced scan, a CTA was performed with the use of contrast injection and the patient was rescanned consecutively with a median delay of 43 s, (range 34–134 s/mean 49 SD ± 19) per protocol. 

CT image analysis was performed on the clinical PACS system, DeepUnity R20 (Dedalus, Florence, Italy) by an experienced interventional radiologist. Unenhanced, arterial and late venous phases in each series were aligned with the use of the built-in 3D co-registration tool (3D Fit). The slice thickness used was 0.8 mm in all cases. Manual corrections were performed where needed. When necessary, full visualization of the clot was achieved with the use of multiplanar reformat (MPR) images. The location of the clot was identified by information gathered from all three phases and the length was measured. 

For dynamic perviousness calculation, three (ROIs) of 1 mm each were placed over the length of the clot as illustrated in Figure 1b. The mean density in HU was first calculated for each ROI separately. After that, the mean density for NCCT, CTA and CTV was calculated by averaging the corresponding three ROIs in the image into a single mean value. This resulted in three mean density values, one each for NCCT, CTA and CTV. The change in density from unenhanced to arterial as well as unenhanced to venous and arterial to venous phases were calculated. From this, four patterns of dynamic perviousness were determined and classified as follows: (1)No contrast uptake (when a change of 5 HU or less over all phases was observed);(2)Late uptake (when a change of 5 HU or less from unenhanced to arterial phase and change of more than 5 HU from unenhanced to late phase were observed);(3)Early uptake with washout (when a change of more than 5 HU from unenhanced to arterial phase and drop of 5 HU or more from arterial to late phase were observed);(4)Early uptake without washout (when a change of more than 5 HU from unenhanced to arterial phase and change of 5 HU or less from arterial to late phase were observed);

Groups 1 and 2 were classified as non-pervious and groups 3 and 4 as pervious thrombi according to the concept of standard perviousness. 

### 2.3. Clot Processing and Histologic Analysis

All retrieved clots were formalin-fixed and paraffin-embedded (FFPE). Sections of two micrometers were cut on a microtome and collected on a TOMO ^®^ slice (Biosystems, Muttenz, Switzerland). Histologic staining was performed with Hematoxylin and Eosin (H&E). Optical slides were then digitalized using a Hamamatsu C9600 slide scanner. Histologic quantification was then performed using Orbit Image Analysis Software (www.orbit.bio, Orbit Image Analysis, Idorsia Ltd., Allschwill, Switzerland) [32] and clots were assessed for relative composition of fibrin/platelets, red blood cells (RBCs) and white blood cells (WBCs). Clots that had an RBC count 15% more than the count of fibrin/platelet were considered RBC-rich, while clots with a 15% higher count of fibrin/platelets over RBC level were considered fibrin/platelet-rich. Clots that did not fit in either category were considered mixed [2,33]. Figure 2 shows an example of a fibrin-rich clot. 

### 2.4. Statistical Analysis

Statistical analysis was performed with the statsmodels library in Python 3.9 version 0.13.5. Normality tests were performed on all the ordinal variables using the Kolmogorov–Smirnov tests and no normally distributed variables were found. Subsequently, statistical analysis was performed using the Kruskal–Wallis tests comparing the uptake categories with the percentages of RBCs, WBCs and fibrin/platelets. The categorical variables of clot types were tested using the chi-squared test and compared to the uptake categories and mTICI scores, all corrected for multiple comparisons using the Benjamini/Hochberg method. The level of statistical significance was set at *p* < 0.05. 

The correlation between uptake and percentages of RBCs, fibrin/platelets and WBC was calculated using Pearson’s correlation analysis with a coefficient (r) of 0.90–1 as very strong; 0.70–0.89 as strong; 0.40–0.69 as moderate; 0.20–0.39 as weak and 0.01–0.19 as a negligible relationship.

## 3. Results

The mean age of the 39 patients was 71 years (SD ± 17) and 19 patients (49%) were female. Occlusions were located at the MCA (*n* = 28, 71.8%—M1-Segment, *n* = 23, M2-segment, *n* = 5) the ICA (*n* = 3, 7.7%), the ICA and MCA (*n* = 5, 12.8%), MCA and ACA (*n* = 1, 2.6%) the basilar (*n* = 1, 2.6%), the PCA, and the P1-Segment (*n* = 1, 2.6%). More than half of the cases were of cardioembolic (TOAST 2, *n* = 20, 51.3%), followed by unknown (TOAST 5, *n* = 16, 41%); large artery atherosclerosis (TOAST 1, *n* = 2, 5.1%) and other determined cause of stroke (TOAST 4, *n* = 1, 2.6%) were the least common causes. The mean NIHSS (National Institutes of Health Stroke Scale) at presentation was 14 (SD ± 4) and 29 patients (74%) received r-tPA before MT. Clots were collected after the thrombectomy procedure regardless of the mTICI score. Successful recanalization of mTICI 2c or 3 was achieved in 30 cases, with the rest being mTICI IIb or 0. In the two cases of mTICI 0, sufficient clot material for analysis was collected, although recanalization was not achieved. An overview of the study population characteristics is shown in Table 1. 

Figure 3 illustrates the Pearson correlation analysis between the structural parameters of the clot (fibrin/platelet to RBC, fibrin/platelet to WBC and RBC to WBC) grouped into four categories according to the measured contrast agent uptake characteristics. Clot with contrast agent uptake without washout as well as clot with washout showed a moderate relationship between RBCs and WBCs (r = 0.45 and r = 0.41, respectively, Figure 3).

Figure 4 compares the measured dynamic perviousness characteristics and the corresponding clot types (fibrin/platelet-rich vs. RBC-rich + mixed). Fibrin/platelet-rich clots showed significantly different contrast uptake patterns when compared to all the rest (RBC-rich plus mixed). The uptake without washout category had significantly fewer fibrin/platelet-rich clots compared to the uptake with washout (*p* = 0.036) and nearly significant fewer fibrin/platelet-rich clots when compared to the no uptake category (*p* = 0.057).

The four uptake categories did not significantly correlate either with the percentage of clot components (Figure 5) or with the final mTICI scores (*p* = 0.61). No statistical significance was found between clot composition and standard perviousness (thrombus attenuation increase from NCCT to CTA).

## 4. Discussion

In the present study, we analyzed the relationship between the novel imaging biomarker dynamic perviousness and the histologic structure of occlusive thrombi in AIS. We found that time-dependent characterization of thrombus permeability to contrast material correlates with thrombus structure. From the four groups of contrast uptake pattern, two categories of early contrast uptake with and without washout would represent more pervious clots. The no uptake and the late uptake groups, where contrast enhancement was present mostly during the venous phase, represent less pervious thrombi according to the definition of standard non-dynamic perviousness [4]. We found that clots with low fibrin/platelet composition show early contrast uptake without washout and clots with high fibrin/platelet composition show early contrast uptake with washout, similar to the no contrast uptake group. When analyzed according to non-dynamic perviousness definitions, we found no significant difference between clot composition types. This raises the potential value of dynamic perviousness classification when it comes to fibrin structure prediction. 

Clot composition and its correlation with stroke etiology as well as the success of endovascular procedures and the effect of thrombolytic treatments has gained interest in the era of MT. A wide range of impactful research has been conducted and strong evidence has been established regarding existing associations. We know that the architecture of cerebral thrombi can be variable [34,35] and that clot composition may influence the effect of thrombolysis, with RBC-rich clot being more prone to lysis [7,36,37]. It has been widely demonstrated that fibrin/platelet-rich clots are more resilient and more difficult to be retrieved compared to RBC-rich clots [33,38], although a recent report showed that increased RBC content is negatively associated with first pass effect [2].

Clot perviousness has risen lately as a possible biomarker that could provide information and enhance our understanding of clot behavior and how that might be related to procedural success and patient outcome [17]. However, there exists a relatively small number of studies that tried to correlate perviousness with histopathological composition. In addition, the reported results are contradictory. Patel et al. [25] and Berndt et al. [39] showed that higher perviousness is correlated with higher percentage of fibrin/platelet composition, while Benson et al. [40] reached the opposite conclusion. 

Lacking clarity of clot type definition might be a reason for the controversial results. Different investigators use different ways to categorize clot into fibrin/platelet-rich and RBC-rich. A lot of histopathologic studies have used macroscopic evaluation without specific thresholds [9,34,41,42], others have used cutoffs of over 60% [43] or more than 15% difference between the main components [2,33,44], and in some, determination of the cutoff was sample specific [40]. Additionally, the way perviousness is defined in the existing literature does not represent the clot’s true permeability characteristics since it does not incorporate flow propagation through the clot. 

In our study, pervious clots show opposing compositions based on their contrast washout pattern. Our results indicate that the way clots interact with contrast might not rely solely on clot composition. Other factors might as well contribute to contrast uptake patterns such as flow dynamics in the occluded vessel and degree of retrograde collateralization, which in turn might affect the concentration of contrast available in proximity to the clot from initial injection to the late venous phase. It might be also reasonable to suggest that rather than relying solely on clot composition, the arrangement of components inside the clot might affect porosity and mechanical properties, which in turn could play an important role in clot behavior. A study investigating platelet-driven contraction, showed that between platelet-contracted clot analogues (PCCs) and non-contracted clot analogues (NCCs), although histologically similar in the RBC and fibrin content, the microstructure of the two clot groups differed significantly [45]. Our proposed categorization of dynamic perviousness might prove to be reflective of such information. This is further supported by the fact that no differences were observed in standard permeability comparisons of clot histological composition.

Although we obtained encouraging results, our study has some limitations. First, the overall patient sample is relatively small and further separation into four different uptake categories makes the number of patients per group even smaller. On the other hand, our study shows statistically significant results, even with such a relatively small number of patients. Second, due to the retrospective nature of the design, we were not able to include consecutive patients. Only patients who had both full imaging data of sufficient quality available and clot histology could be included, which may lead to potential bias. Third, the vast majority of patients had cardioembolic or unknown origin of stroke and only a few patients were considered to have stroke of large artery atherosclerotic origin, which is one of the main causes of thromboembolic events [46,47,48]. Therefore, the patient group of this study does not cover each possible clot origin in AIS. Fourth, because of its retrospective design, CT protocols did not use a constant time delay between CTA and CTV. In the acute treatment of AIS, it is a challenge to maintain precise time intervals between CT acquisitions. To test its influence on the statistical analysis, the time delay between CTA and CTV was analyzed for the parameter groups and no significant differences were found. Nevertheless, the varying time delay between CTA and CTV introduced a potential bias into our analysis. Fifth, we used standard CT protocols instead of specific protocols tailored for the research question of this study. This is because a comprehensive and time-consuming CT protocol would not be ethical and we need to rely on the data that are provided by the routine scans. On the other hand, the here used standard CT protocols are available in almost every stroke unit, which increases the value of the presented method. Nevertheless, it would be of great interest to correlate in vivo thrombus structure with more comprehensive imaging modalities, such as MRI.

## 5. Conclusions

This preliminary study indicates that the concept of dynamic perviousness may increase the potential for clot characterization using pre-interventional CT imaging. Contrast uptake with different patterns of contrast washout showed significant differences in the likelihood of fibrin-rich clots. Future studies with a higher number of patients will help clarify the relationship between dynamic perviousness and clot histology/structure.

## Figures and Tables

**Figure 1 diagnostics-14-01387-f001:**
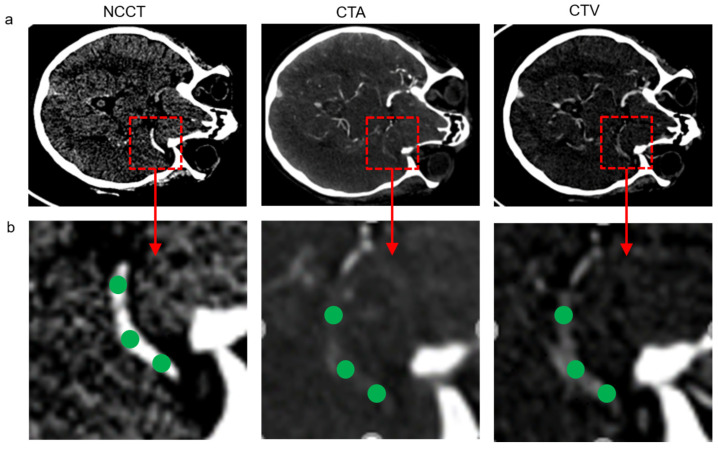
(**a**) Representative non-contrast CT (NCCT), CT angiography (CTA) and late venous phase CT (CTV) images of our clinical AIS protocol. (**b**) Illustration of spherical ROIs placed over the clot in all three images for dynamic perviousness calculation.

**Figure 2 diagnostics-14-01387-f002:**
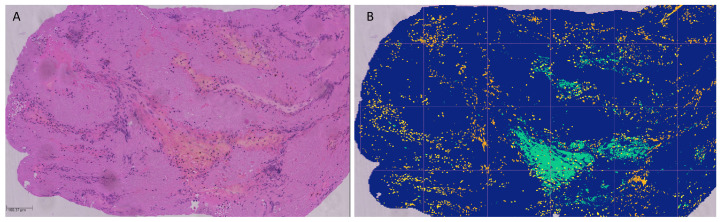
HE staining of a fibrin-rich clot (**A**) and the automated quantification map by Orbit^®^ (**B**), with blue coding for fibrin platelets, green for red blood cells and yellow for white blood cells, representing a total of 86% fibrin content.

**Figure 3 diagnostics-14-01387-f003:**
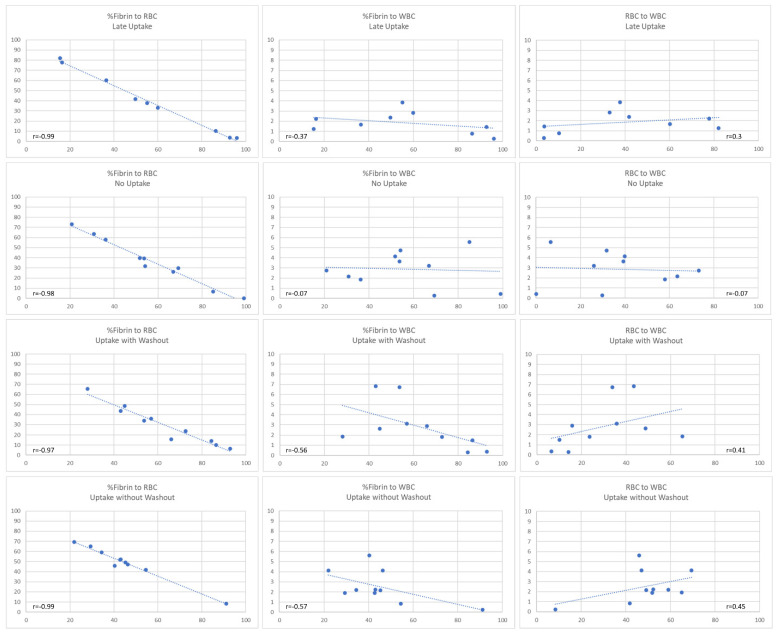
Graphic representation of Pearson correlations between fibrin/platelets (labeled as fibrin in the graph), red blood cells (RBCs) and white blood cells (WBCs). Correlations between fibrin/platelet to RBC, fibrin/platelet to WBC and RBC to WBC grouped into four categories according to the measured contrast agent uptake characteristics are shown. A moderate relationship between RBCs and WBCs in the uptake without washout (r = 0.45) and the uptake with washout category (r = 0.41) is seen. The Pearson correlation coefficient (r) measures the strength and direction of a linear relationship between the two variables.

**Figure 4 diagnostics-14-01387-f004:**
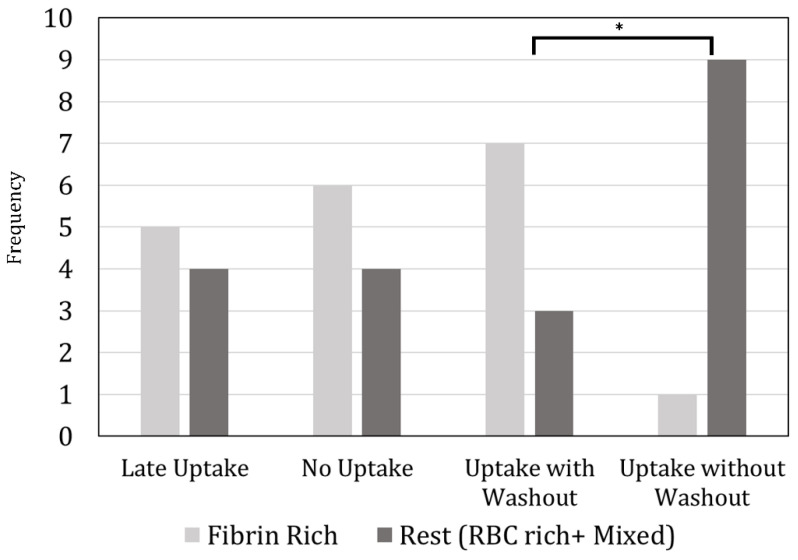
Comparison between contrast uptake categories and clot types. Significance of *p* = 0.036 was found only for the fibrin/platelet-rich clots between uptake with and uptake without washout. The uptake with washout category showed significantly more fibrin/platelet-rich clots compared to the uptake without washout category. Comparison between the other uptake categories did not show any significant differences of clot components. Asterisk placed above categories where statistical significance was found.

**Figure 5 diagnostics-14-01387-f005:**
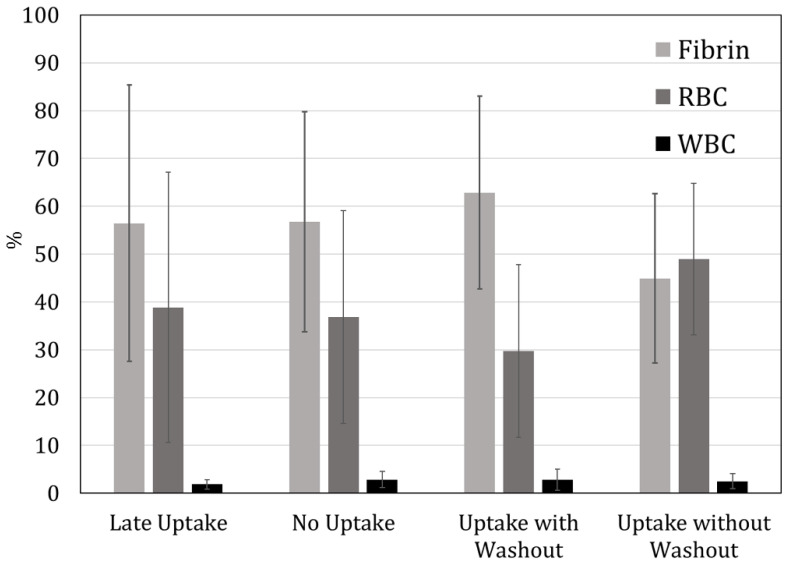
Comparison of percentages of clot components between the different contrast uptake categories. No significance was found.

**Table 1 diagnostics-14-01387-t001:** Characteristics of study population.

**Age, Mean Years ± SD**	**71 ± 17**
**Gender, *n* (%)**	
Male	20 (51%)
Female	19 (49%)
**Stroke Etiology, *n* (%)**	
Large vessel disease	2 (5.1%)
Cardioembolic	20 (51.3%)
Other determined cause	1 (2.6%)
Unknown	16 (41%)
**Site of occlusion, *n* (%)**	
MCA—M1 segment	23 (59%)
MCA—M2 segment	5 (12.8%)
ICA	3 (7.7%)
ICA + MCA	5 (12.8%)
MCA + ACA	1 (2.6%)
Basilar	1 (2.6%)
PCA—P1 segment	1 (2.6%)
**NIHSS, mean ± SD**	14 ± 4
**r-tPA, *n* (%)**	29 (74%)
**mTICI score, *n* (%)**	
2c, 3	30 (76.9%)
2b	7 (17.9%)
0	2 (5.1%)

MCA: middle cerebral artery, ACA: anterior cerebral artery, ICA: internal carotid artery, PCA: posterior cerebral artery, NIHSS: National Institutes of Health Stroke Scale, r-tPA: recombinant tissue plasminogen activator, mTICI: modified thrombolysis in cerebral infraction.

## Data Availability

The data presented in this study are available on request from the corresponding author. The data are not publicly available due to data protection regulations.

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
