# Peer review of "Dynamic Perviousness Has Predictive Value for Clot Fibrin Content in Acute Ischemic Stroke"

_diagnostics, 2024, doi:10.3390/diagnostics14131387_

Round 1
Reviewer 1 Report
Comments and Suggestions for Authors
The present manuscript titled “ Dynamic perviousness has predictive value for clot fibrin content in acute ischemic stroke” led by Vania et al show fibrin content is largely accumulated in the clot after performing the thrombectomy in LVO. Though the present work is novel, the following corrections need to be addressed by authors.
Comments:
· Remove full stop from the title.
· Provide space between words in the entire abstract.
· Line 23: ICA occlusion percentage is missing.
· Line 34: Mechanical, use lower case for M.
· Line 112 to 118. There is a repetition of serial numbers. Remove it.
· Authors need to discuss the inclusion and exclusion criteria of the patients form the study.
· The representative images of occlusion must be placed to understand the extent of clot.
· How authors assessed the blood cells in the clot? Provide the H and E images of clot. Further, the number of samples were used for histology studies is missing. Provide these details
· There are several punctuation errors (comma, full stop, and spaces) thorough out the manuscript. Correct them.
· Discuss the limitations of the study.
Comments on the Quality of English LanguageGrammar and punctuations to be corrected throughout the manuscript
Reviewer 2 Report
Comments and Suggestions for Authors
The authors of the manuscript describe an original clinical study aimed at solving the urgent task of developing neuroimaging methods and X-ray markers of a blood clot condition based on the analysis of data on its contrast and structure, which makes it possible to predict the effectiveness of treatment.
Despite the limitations of the study (retrospective study design, small sample size), the data obtained are of high importance for practical healthcare. The results shown by the authors can make a significant contribution to the development of methods for diagnosing and predicting outcomes based on CT diagnostics with dynamic blood clot permeability. Therefore, the conducted research may be relevant for publication in the journal Diagnostics.
During the review of the manuscript, several points were found that can be improved:
1. In the "Introduction" section, it is desirable to additionally describe the background of the introduction of the concept of dynamic permeability and its biological/technological essence.
2. In the "Discussion" section, you can (although not necessarily) add information about the prospects for the development of the methodology. For example, the possibility of supplementing the analysis of neuroimaging data with artificial intelligence methods.
Reviewer 3 Report
Comments and Suggestions for Authors
Dear authors,
I would like to congratulate you on the article you submitted to the journal. You address a particularly significant and original topic and handle it clearly. Please allow me to make some comments:
Line 69: I propose to avoid the reference to your results in that point.
Line 80: I propose to change the phrase “see bellow” …
Line 149: Results : I would suggest a Figure for demographics instead of a description
Thank you
Reviewer 4 Report
Comments and Suggestions for Authors
Dear Authors,
Thank you for the opportunity to review your manuscript titled "Dynamic perviousness has predictive value for clot fibrin content in acute ischemic stroke." Your research addresses an important topic in the field of stroke imaging and offers valuable insights. Below, I provide detailed constructive feedback to enhance the quality and impact of your manuscript.
1. Introduction
Comments:
l Background and Relevance: The introduction provides a comprehensive background on the significance of clot characteristics in acute ischemic stroke and their impact on revascularization outcomes. However, the flow of information can be improved for better readability.
l Clarity and Coherence: Some sentences are overly complex and may benefit from simplification. For example, the sentence "Dynamic perviousness is a novel imaging biomarker, whith clot density measurements at multiple timepoints to allow longer contrast to thrombus interaction" should be corrected to "Dynamic perviousness is a novel imaging biomarker that measures clot density at multiple time points, allowing for longer contrast interaction with the thrombus."
2. Research Design
Comments:
l Appropriateness: The research design is generally appropriate for the study’s objectives. However, providing more detail on patient selection criteria and the rationale for choosing specific imaging protocols would strengthen the design.
l Examples: Explain why three-phase CT imaging was chosen and how it compares to other imaging methods.
3. Methods
Comments:
l Description: The methods are adequately described but can be made more concise and clearer. Ensure consistency in terminology and detail the steps logically.
l Examples: The explanation of the imaging protocol and the calculation of dynamic perviousness are well-articulated but can be streamlined. For instance, break down the imaging phases and density calculations into separate, clear steps.
4. Results
Comments:
l Presentation: The results section presents data effectively but could benefit from clearer visual aids and explanations. Consider reorganizing tables and figures for better comprehension.
l Clarity: Enhance the clarity by explaining statistical results more plainly. For example, describe the significance of p-values in layman's terms to ensure all readers understand their importance.
5. Discussion
Comments:
l Interpretation: The discussion interprets the results well, but some points are repetitive. Streamline the discussion to focus on key findings and their implications.
l Flow: Ensure smooth transitions between paragraphs. For example, connect the findings on dynamic perviousness directly to clinical implications without unnecessary detours.
6. English Language Quality
Comments:
l Grammar and Syntax: There are several grammatical errors and awkward phrasings throughout the manuscript. Sentences such as "By etiology, more than half were of cardioembolic origin" can be rephrased for clarity: "More than half of the cases were of cardioembolic origin."
l Consistency: Maintain consistency in terminology. For instance, standardize terms like "fibrin-rich" and "fibrin/platelet-rich" throughout the manuscript.
l Typographical Errors: Correct typographical errors and ensure proper punctuation to avoid run-on sentences and improve readability.
l Example: "The use of only two imaging time points (NCCT and CTA), which neglects the dynamic nature of contrast agent penetration by under sampling the time-resolved contrast agent uptake curve, can lead to a massive underestimation of thrombus perviousness." This can be simplified to: "Using only two imaging time points (NCCT and CTA) underestimates thrombus perviousness by neglecting the dynamic nature of contrast agent penetration."
Recommendations for Improvement
l Professional Editing: Consider having the manuscript professionally edited by a native English speaker with experience in scientific writing to correct grammatical and syntactical errors.
l Peer Review: Have colleagues or peers review the manuscript for readability and clarity. Their feedback can provide valuable insights from a fresh perspective.
l Use of Writing Tools: Utilize writing tools such as Grammarly or Hemingway to identify and correct language issues.
l Enhancing Visuals: Improve the clarity of tables and figures. Ensure they are easy to interpret and directly support the text.
l Streamlining Content: Remove redundant information and ensure each section transitions smoothly to maintain the reader’s engagement.
By addressing these points, you can significantly enhance the quality and impact of your manuscript. I look forward to seeing the revised version and the potential improvements in clarity and readability.
Best regards,
Tom
The Reviewer
Comments on the Quality of English Language1. Grammar and Syntax:
l Examples of Issues:
n "Dynamic perviousness is a novel imaging biomarker, whith clot density measurements at multiple timepoints to allow longer contrast to thrombus interaction." ("whith" should be "with")
n "By etiology, more than half were of cardioembolic origin." (Consider rephrasing for clarity: "Etiology-wise, more than half of the cases were of cardioembolic origin.")
l Recommendation: The manuscript contains several grammatical errors and awkward phrasings. A thorough review of grammar and syntax is necessary to improve readability and comprehension.
2. Consistency:
l Examples of Issues:
n The term "dynamic perviousness" is used frequently, but sometimes without clear definition or context in different parts of the manuscript.
n Variations in terminology (e.g., "fibrin-rich" vs. "fibrin/platelet-rich") should be standardized throughout the text.
l Recommendation: Ensure consistent use of terms and phrases. Standardize terminology to avoid confusion.
3. Clarity and Precision:
l Examples of Issues:
n "The use of only two imaging time points (NCCT and CTA), which neglects the dynamic nature of contrast agent penetration by under sampling the time-resolved contrast agent uptake curve, can lead to a massive underestimation of thrombus perviousness."
(The sentence is too long and complex, making it difficult to follow.)
l Recommendation: Break down long sentences into shorter, clearer ones. Aim for precision and clarity in conveying complex ideas.
4. Flow and Coherence:
l Examples of Issues:
n Some sections of the manuscript, particularly the introduction and discussion, lack smooth transitions between paragraphs and ideas.
l Recommendation: Improve the logical flow of ideas by adding transitional phrases and ensuring each paragraph leads naturally to the next.
5. Technical Terminology:
l Examples of Issues:
n Technical terms are sometimes introduced without adequate explanation for a broad audience.
l Recommendation: When introducing technical terms or concepts, provide brief explanations or definitions to ensure all readers can follow the discussion.
6. Typographical Errors:
l Examples of Issues:
n "This important clot characteristic is related to porosity and to the shape and connectivity of the pores created between fibrin filaments, trapped platelets and red blood cells (5-7)."
n Ensure proper use of punctuation, such as commas and periods, to avoid run-on sentences and improve readability.
l Recommendation: Proofread the manuscript to correct typographical errors and ensure proper punctuation.
Suggested Actions:
1. Professional Editing:
Consider having the manuscript professionally edited by a native English speaker with experience in scientific writing.
2. Peer Review:
Have colleagues or peers review the manuscript for readability and clarity. They can provide valuable feedback from a fresh perspective.
3. Use of Writing Tools:
Utilize writing tools and software such as Grammarly or Hemingway to identify and correct language issues.
Improving the quality of the English language in the manuscript will enhance its readability and ensure that the scientific content is conveyed effectively to the readers.
Round 2
Reviewer 1 Report
Comments and Suggestions for Authors
There are several punctuation errors in the manuscript. Remove unnecessary comma and full stop. Remove or add space.
Comments on the Quality of English LanguageThere are several punctuation errors in the manuscript. Remove unnecessary comma and full stop. Remove or add space.
Reviewer 3 Report
Comments and Suggestions for Authors
Dear authors, thank you very much for the changes you have made. I have no further comments on your article.
Reviewer 4 Report
Comments and Suggestions for Authors
Dear Authors,
Thank you for submitting the revised version of your manuscript titled "Dynamic perviousness has predictive value for clot fibrin content in acute ischemic stroke." We appreciate the effort and time you have invested in addressing the reviewers' comments. Based on a detailed comparison with the original version and the author response letter, we have the following comments and suggestions to help further improve the manuscript and ensure it is ready for publication:
1. Introduction:
l Background and Flow of Information:
The revised introduction provides a comprehensive background on the significance of clot characteristics in acute ischemic stroke and their impact on revascularization outcomes. The flow of information has been improved, making it more readable and coherent. However, some sentences could still be simplified to enhance readability. For example, the sentence "Dynamic perviousness is a novel imaging biomarker that measures clot density at multiple time points, allowing for longer contrast interaction with the thrombus" is much clearer now. We suggest continuing this approach throughout the introduction.
l Relevant References:
The introduction includes relevant references that support the study's context. Ensure that all key studies are cited to provide a robust background.
2. Research Design:
The research design is generally appropriate for the study’s objectives. The inclusion of additional detail on patient selection criteria and the rationale for choosing specific imaging protocols has strengthened the design. However, further clarification on the limitations of using a retrospective design and the potential biases introduced by patient selection criteria would be beneficial. Consider elaborating on these aspects in the "Limitations" section.
3. Methods:
The methods section is adequately described and the steps are detailed logically. The restructuring and inclusion of Figure 1, which illustrates the imaging protocol and ROIs for dynamic perviousness calculation, have improved clarity. However, ensure consistency in terminology throughout the methods section. For instance, the term "dynamic perviousness" should be used uniformly to avoid confusion.
4. Results:
The results are clearly presented, but the section could benefit from clearer visual aids and explanations. The reorganization of tables and figures has improved comprehension, but further enhancements are recommended. For example, explaining the significance of p-values in layman's terms would make the results more accessible to a broader audience. Additionally, consider adding more descriptive captions to figures to help readers understand the data at a glance.
5. Discussion:
The discussion interprets the results well, but some points remain repetitive. The streamlining efforts are noticeable, but further reduction of redundant information is recommended. Ensure smooth transitions between paragraphs and focus on key findings and their implications. Highlight how the findings of dynamic perviousness can influence clinical practice and future research.
6. Grammar and Syntax:
While the grammatical errors and awkward phrasings have been largely addressed, minor editing is still required. For example, sentences like "More than half of the cases were of cardioembolic origin" are clearer now. However, ensure that similar improvements are applied consistently throughout the manuscript. Correct any remaining typographical errors and ensure proper punctuation to improve readability further.
7. Originality and Novelty:
The study presents a novel approach by investigating dynamic perviousness and its correlation with clot histologic composition. This originality is a strong point of the manuscript and adds significant value to the existing literature.
8. Significance of Content:
The content is significant and provides valuable insights into the predictive value of dynamic perviousness in acute ischemic stroke. The findings have the potential to influence clinical practice and future research in stroke management.
9. Overall Merit:
The manuscript is of high overall merit, and with the minor suggested revisions, it will be ready for publication. The quality of the study, the significance of the findings, and the clarity of presentation are commendable.
Conclusion:
The revision has addressed the majority of the concerns and comments from the reviewer effectively. With minor additional improvements in clarity, consistency, and readability, the manuscript will be ready for publication. We commend the authors for their efforts and look forward to seeing this valuable contribution published.
Best regards,
Tom
The Reviewer
Comments on the Quality of English LanguageRegarding the quality of the English language in the revised manuscript, the overall readability and clarity have improved significantly. However, there are still areas that require attention to ensure the manuscript meets the high standards of academic writing. Here are some specific and detailed comments:
1. Sentence Structure and Clarity:
l The manuscript generally maintains clear and coherent sentence structures, but some sentences are still complex and could be simplified for better readability. For instance, in the introduction, the sentence "Dynamic perviousness is a novel imaging biomarker that measures clot density at multiple time points, allowing for longer contrast interaction with the thrombus" is clear, but similar clarity should be applied throughout the manuscript.
l Example of a complex sentence that can be simplified: "The use of only two imaging time points (NCCT and CTA), which neglects the dynamic nature of contrast agent penetration by under sampling the time-resolved contrast agent uptake curve, can lead to a massive underestimation of thrombus perviousness." Simplified: "Using only two imaging time points (NCCT and CTA) underestimates thrombus perviousness by neglecting the dynamic nature of contrast agent penetration."
2. Grammar and Syntax:
l While the majority of grammatical errors have been corrected, minor issues still persist. For example, ensure subject-verb agreement and proper use of articles.
l Example: "By etiology, more than half were of cardioembolic origin." can be rephrased to "More than half of the cases were of cardioembolic origin."
3. Consistency in Terminology:
l Maintain consistency in the use of terms throughout the manuscript. Terms like "fibrin-rich" and "fibrin/platelet-rich" should be standardized. Inconsistencies can confuse readers and detract from the overall clarity of the paper.
l Example: If you refer to "fibrin-rich clots" in one section, ensure that the same term is used throughout, rather than alternating with "fibrin/platelet-rich clots."
4. Typographical Errors:
l Correct any remaining typographical errors to ensure a polished final manuscript. This includes issues like missing commas, incorrect spacing, and capitalization errors.
l Example: Ensure consistent use of capital letters at the beginning of sentences and proper punctuation throughout.
5. Flow and Coherence:
l Ensure smooth transitions between sentences and paragraphs to maintain a logical flow of ideas. This is particularly important in the discussion section where you interpret the results.
l Example: Use transitional phrases such as "Furthermore," "Additionally," and "In contrast," to connect ideas and findings.
6. Passive vs. Active Voice:
l While passive voice is often used in scientific writing, consider using active voice where appropriate to make sentences more direct and engaging.
l Example: "The patients were selected based on the dynamic perviousness classification" can be rephrased to "We selected the patients based on the dynamic perviousness classification."
7. Technical Jargon:
l While technical terms are necessary, ensure that their use is appropriate and that they are clearly defined when first introduced. This will help readers who may not be familiar with the specific jargon.
l Example: Clearly define "dynamic perviousness" and any other specialized terms early in the manuscript.
Conclusion:
The manuscript's English language quality has improved but requires minor editing to address the issues mentioned above. Simplifying complex sentences, ensuring grammatical accuracy, maintaining consistency, and improving the flow will enhance the readability and overall quality of the manuscript.
